# Electrocatalytic Reduction of O₂ by ITO-IrOx: Implication for Dissolved Oxygen Sensor in the Alkaline Medium

Munira Siddika [1,†], Md. Mahmudul Hasan [2,*,†], Tahamida A. Oyshi [1] and Mohammad A. Hasnat [1,*]

1 Electrochemistry & Catalysis Research Laboratory (ECRL), Department of Chemistry, School of Physical Sciences, Shahjalal University of Science and Technology, Sylhet 3114, Bangladesh
2 Research Organization for Nano and Life Innovation, Waseda University, Shinjuku, Tokyo 162-0041, Japan
* Correspondence: hasan@aoni.waseda.jp (M.M.H.); mah-che@sust.edu (M.A.H.)
† These authors contributed equally to this work.

**Abstract:** Water pollution has badly affected human health, aquatic life, and the ecosystem. The purity of surface water can be measured in terms of dissolved oxygen (DO) measurements. Hence, it is desirable to have a portable and simple-to-use dissolved oxygen sensor. One possible remedy is an electrochemical sensor. Thus, we proposed an ITO-IrOx electrocatalyst for an effective and interference-free DO sensor utilizing the principle of oxygen reduction reaction (ORR). The ITO-IrOx was characterized using cyclic voltammetry (CV), scanning electron microscopy (SEM), electro-chemical impedance spectrometry (EIS), X-ray photoelectron spectroscopy (XPS), and reflectance spectroscopy-based techniques. Reflectance spectra of the ITO-IrOx electrode showed the photoresist capability. The EIS spectra revealed lower charge transfer resistance for the ITO-IrOx electrode in ORR. The IrOx film on ITO exhibited a quick (one electron, $\alpha = 1.00$), and reversible electron transfer mechanism. The electrode demonstrated high stability for oxygen sensing, having a limit of detection (LOD) of 0.49 ppm and interference-free from some common ions (nitrate, sulphate, chloride etc.) found in water.

**Keywords:** Iridium oxide; oxygen reduction reaction; dissolved oxygen; sensitivity; stability





## 1. Introduction

A variety of anthropogenetic activities are threatening the cleanliness of the aqueous systems over the globe. Water is continuously being contaminated by people. While contaminants are present in water, different water parameters have deviated from their standard values [1–3]. Three parameters are commonly used to indicate the quality of surface water. For instance, dissolved oxygen (amount of oxygen present in surface water, DO), biological oxygen demand (number of organic pollutants in water: the amount of oxygen required to degrade organic pollutants by biological means), and chemical oxygen demand (the quantity of organic and inorganic pollutants in surface water: the amount of oxygen required to break down the pollutants by use of chemicals) [4–7].

The quality of water can be confirmed by monitoring dissolved oxygen in the water body. Hence, scientists have paid considerable effort for developing DO measurement methods. The most widely applied methods for DO measurement are Winkler, electrochemical, and photometry [8–11]. The detection of DO may be obtained quickly and conveniently using an electrochemical approach. In the electrochemical process, DO gets directly reduced at the cathode and gives corresponding reduction current which is calibrated to determine the amount of dissolved oxygen present. The most classical electrodes which are often used for DO sensing are the Clark electrode and its modified forms [12,13]. Several other electrodes have been reported for the oxygen sensor, particularly polymer-based cathode modification [14–16]. Yun Zhao et al., used a composite material of fluorinated xerogel with platinum porphyrin dye for the oxygen sensor [14]. Meng Li et al., reported an oxygen sensor based on a polyaniline-modified gold surface [15]. Moreover, most reported oxygen

sensors have some drawbacks such as interference of other ions and stability of the electrode. As a result, scientists are improving the electrochemical DO measurement technique by developing more sensitive, specific, interference-free, affordable, and stable electrodes.

Electrocatalysts based on transition metals can demonstrate excellent catalytic activity [17–19]. Iridium is a d-block transition metal of group nine in the periodic table. It has a high corrosion resistivity. Among all the transition metals in the periodic table, compounds formed by iridium display oxidation states from −3 to +9, resulting in various applications [20]. Additionally, recent advancements in the area of nanomaterials have multiplied its use. Researchers have prepared iridium-based nanoparticles, particularly, IrOx NPs and reported their application in different areas including ORR [21–26].

Indium tin oxide (ITO) is a very stable, durable substrate that may be applied to electrochemical sensors. ITO is a quartz substance that has been widely used as a substrate for catalyst deposition because it increases the electrical conductivity, has a large band gap, and a low cost. It is the combination of $In_2O_3$ and Sn, where tin substitutes for the indium site are incorporated as dopants in the lattice of $In_2O_3$. The ITO glass of a band gap higher than 3 eV is extremely transmitting in the visible region and thus is desired in most applications. In addition, a wide band gap is attractive for the use of the device for high-temperature purposes. Several articles have been published describing the combination of IrOx catalyst with ITO for different applications [27,28]. However, no article is published yet which deals with oxygen reduction reaction (ORR) over ITO-IrOx.

Note that the ORR process involves a number of simple steps involving several reaction intermediates. When molecular oxygen is reduced at the cathode, it occurs either via route 1, where $H_2O_2$ is the final product after accepting up two electrons (2e-transfer process) or via route 2, where $H_2O$ is the final product after accepting four electrons (4e-transfer process) as defined below.

$$\text{Route 1: } O_2 + 2H^+ + 2e^- \rightarrow H_2O_2$$

$$\text{Route 2: } O_2 + 2H^+ + 2e^- \rightarrow H_2O_2$$

$$H_2O_2 + 2H^+ + 2e^- \rightarrow 2H_2O$$

$$\text{Overall: } O_2 + 4H^+ + 4e^- \rightarrow 2H_2O$$

In-situ hydrogen peroxide creation opens the door for dye degradation and wastewater treatment by generating highly active free radicals during the process. Hydrogen peroxide is also an oxidant and can be used in a fuel cell. Route 2 is often used in the half-cell of any type of fuel cell. Yet, the electrolytes and electrode materials have a significant impact on the ORR pathways. Hence, the majority of articles published on ORR have researched the kinetics and number of electron transfers that are associated with ORR. Based on the number of electron transfers, catalysts are proposed for suitable applications such as wastewater treatment and/or fuel cell applications. For instance, Fernandes et al., prepared four different catalysts for ORR by placing phosphotungsate on four different carbon materials such as graphene flakes, single-walled carbon nanotube, graphene doped with nitrogen, and carbon nanotube doped nitrogen [29]. They reported the comparative kinetics of four catalysts towards oxygen reduction reaction in an alkaline medium. As three of them proceed four electron transfer process. Hence, the prepared electrodes were suggested for fuel cell application. Yanyan Sun et al., described the synthesis procedure of nitrogen and phosphorus dual-doped carbon nanosheets (NPCNS) [30]. They prepared NPCNS by pyrolysis of chitosan and phytic acid. Because of the unique 2D nanostructure, the synergistic action of the nitrogen and phosphorus dopant in NPCNS showed strong electrochemical ORR activity and selectivity toward $H_2O_2$ generation in an alkaline medium. The high production of hydrogen peroxide was the main target of their research. Islam et al., claimed that electroless deposition of Au does not occur on bare GCE [31]. Thus, in order to prepare GCE for the deposition of Gold NPs, GC was pretreated before the deposition of gold. They were able to confirm through the analysis of hydrodynamic

voltammograms that the electrochemical reduction of oxygen over GCE-AuNPs produced peroxide. Hence, they used this to break down methylene blue using the electro-fenton process. However, although a number of articles have been published recently on ORR, none of them reports molecular oxygen sensing for environmental monitoring. It is crucial since people frequently contaminate water. Recently, there was a severe pandemic, and particularly in developing nations, medical waste was dumped into rivers and the ocean. To maintain the survival of aquatic life, it is therefore vital to monitor the DO of water bodies using a reliable sensor.

Thus, in this article, we have reported ORR catalysis over the ITO-IrOx electrode to fabricate an oxygen sensor. We chose an alkaline medium for our experiment because most industrial effluents are alkaline in nature. Congruently, we immobilized IrOx nanoparticles on the ITO surface (ITO-IrOx) by electrodeposition and employed them for DO sensing using molecular ORR in the alkaline medium. This study also reports the stability of the electrode, ion interference, and kinetic analysis of the ORR.

## 2. Materials and Methods

### 2.1. Chemicals

Sodium hydroxide (NaOH), Potassium hexachloroiridate ($K_2IrCl_6$), potassium chloride (KCl), hydrochloric acid (HCl), and ethanol ($C_2H_5OH$) were purchased from Sigma Aldrich. The supplied source of $N_2$ and $O_2$ was Lindy, Bangladesh. All of the chemicals used in this study were analytical grade.

### 2.2. Preparation of Precursor Solution for Electrode Modification ($Ir_2O_3 \cdot nH_2O$ Colloid)

A precursor solution for electrode modification ($Ir_2O_3 \cdot nH_2O$) was prepared following the literature [32]. At first, solid $K_2IrCl_6$ (0.13 g) was dissolved in 0.1 M HCl (10 mL). After that, $C_2H_5OH$ (1.5 mL) was also added to the mixture. The subsequent solution was then boiled at ~100 °C under stirred conditions. Water was added to maintain the volume of the solution. After 2 h of constant heating, at every 10 min interval, 100 μL of $C_2H_5OH$ was added. In the interim time, the solution changed its colour to pale blue. Ethanol was added until the solution had no colour effect. Then the excess ethanol was removed by boiling the solution for 1 hr. After that, the final solution was cooled to 25 °C and the solution was made alkaline (pH~12) by adding the required amount of NaOH (0.5 M). This solution was utilized to modify electrodes ($Ir_2O_3 \cdot nH_2O$ colloidal suspension) at pH~12 under an $N_2$ blanket.

### 2.3. Fabrication of Electrode and Electrochemical Measurements

Electrochemical workstations such as CHI 660 (CHI Instruments, Austin, TX, USA) and Autolab 128 N (Herisau, The Netherlands) were used for the electrochemical studies and IrOx immobilization on the ITO surface. A glass cell with three electrodes was used for all of the electrochemical experiments. A platinum (Pt) wire, an ITO electrode with a diameter of 3 mm and the Ag/AgCl (sat. KCl) electrode served as the counter electrode, working electrode, and reference electrode, respectively.

Before the deposition, the ITO glass was cleaned by sonication with ethanol. Then the cleaned ITO surface was dried for 2 h in an oven (70 °C). Then, 10 mL of prepared colloidal suspension was taken in a cell fitted with ITO glass for electrodeposition. The potential window for electrodeposition was used from 0 to +1.0 V vs. Ag/AgCl (sat. KCl) at a scan rate of 100 mV s$^{-1}$ for up to five cycles (Figure 1). The prepared electrode was cleaned using distilled water and allowed to air dry. The modified electrode ITO-IrOx was then employed as a working electrode for next investigations (characterization, ORR catalysis, oxygen sensing, etc.).

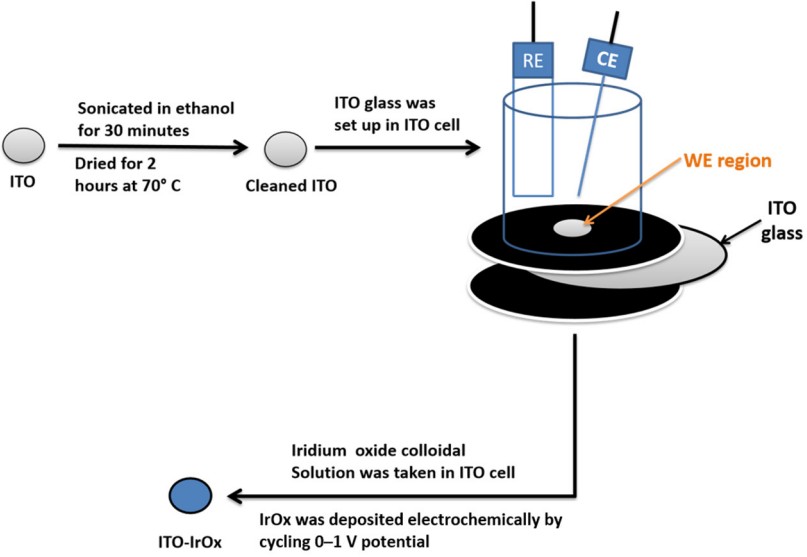

**Figure 1.** Schematic diagram of ITO-IrOx fabrication.

*2.4. Surface Analysis*

The scanning electron microscope (SEM, JEOL, JSM-7600F, Tokyo, Japan) was used to investigate the surface morphology of ITO-IrOx surface. The X-ray photoelectron spectroscopy (XPS) with a K-$\alpha$1 spectrometer (Thermo Scientific, K-$\alpha$1 1066, Austin, TX, USA) and an excitation radiation source (A1 K-$\alpha$1, Beam spot size = 300.0 µm) was employed to investigate the elemental properties of ITO-IrOx. The semiconducting property of IrOx film on ITO glass was studied by Reflectance spectroscopy (Avantes, Apeldoorn, The Netherlands).

### 3. Results and Discussion

*3.1. Spectroscopic Characterization*

Figure 2A,B represented the SEM images of ITO and ITO-IrOx, respectively. It can be seen from the SEM images that the particles of iridium oxide on ITO appeared as clusters having no regular shape. The XPS of Ir 4f for ITO-IrOx is displayed in Figure 2C. Two high-intensity peaks at binding energy 64.7 eV and 61.9 eV revealed the existence of Ir $4f_{5/2}$ and Ir $4f_{7/2}$ respectively [33–35]. In the case of Ir $4f_{7/2}$ and Ir $4f_{5/2}$, the binding energy splitting of the spin-orbit doublet is 3.0 eV. Here, we found a binding energy difference of 2.8 eV with an intensity ratio of 4:3. The result is consistent with the literature [34,36,37]. Two minor peaks at binding energy 65.7 eV and 63.0 eV have been ascribed to the existence of Ir (IV) and Ir (V) species, respectively [30,32,38]. Figure 2D shows the XPS of O 1s. Three different forms of oxygen can be seen here. The peaks at high binding energy (532.5 eV) and intermediate binding energy (531.5 eV) could be caused by the water oxygen atom and OH$^-$, respectively [38,39]. The lower binding energy peak (529.8 eV) is attributed to IrOx [32,40,41]. The existence of different oxidation states of Ir is mainly responsible for catalyzing various electrochemical reactions.

However, to obtain band gap energy, reflectance spectra of ITO-IrOx and plain ITO electrodes were recorded. Band gap energy was determined by using the Kubelka-Munk theory as follows (Equations (1) and (2))

$$\alpha = \frac{A}{h\vartheta}\left(h\vartheta - E_g\right)^n \tag{1}$$

Here, *A* is a constant, $\alpha$ = absorption coefficient and *hv* = photon energy. The direct band gap was obtained from the relation

$$\alpha h\vartheta = A\left(h\vartheta - E_g\right)^{\frac{1}{2}} \tag{2}$$

From the reflectance spectra (Figure 3A) of ITO-IrOx, it is seen that numerous bands reflected the formation of a uniform film of IrOx on ITO. By extrapolation of the linear portion on the abscissa, the band gap was calculated. The band gap energy of ITO-IrOx (4.2 eV) was discovered to be more than plain ITO (3.4 eV). This means that, the modified electrode would not be photosensitive. In other words, the developed electrode could be employed without any interference from sunlight.

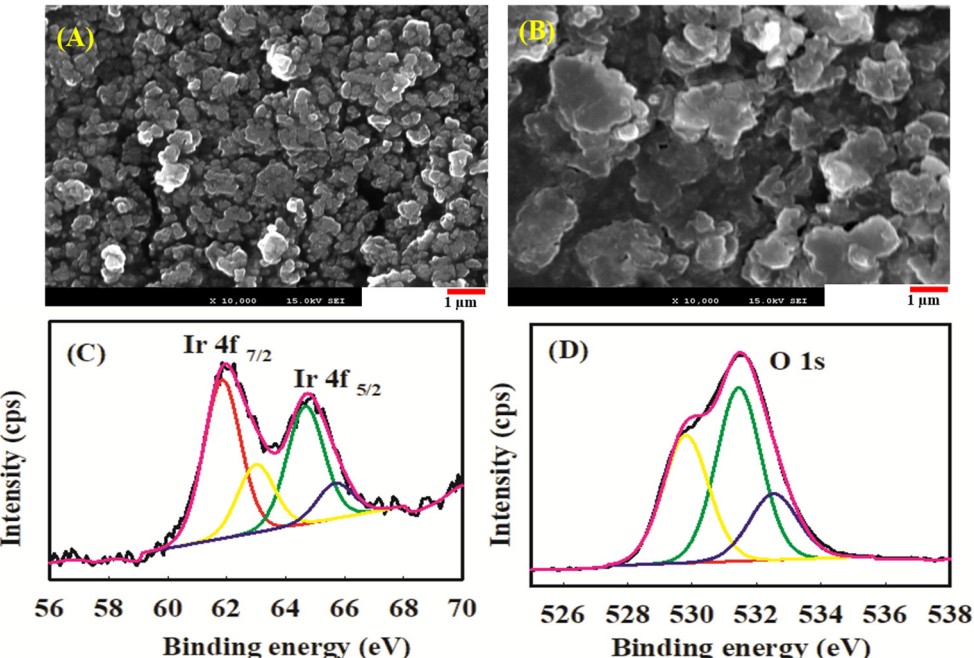

**Figure 2.** SEM image of (**A**) ITO and (**B**) ITO-IrOx (scale bar = 1 μm) and XPS spectra of (**C**) Ir 4f and (**D**) O 1s for ITO-IrOx surface.

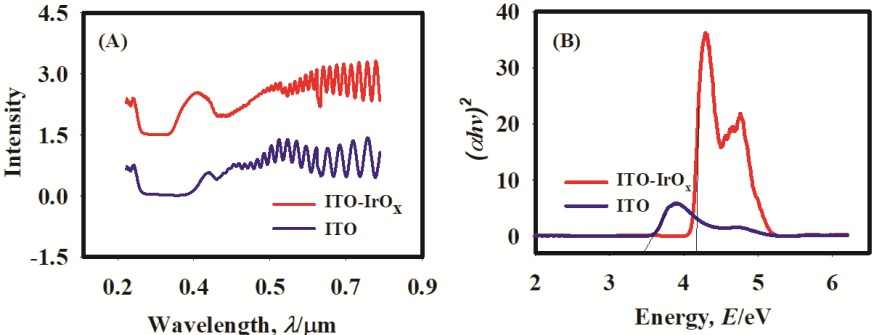

**Figure 3.** (**A**) The Reflectance spectra of ITO and ITO-IrOx electrode as a function of wavelengths. (**B**) Tauc plots for calculating band gap energy (intersection of the black line with the E axis represents the band gap energy).

### 3.2. Voltammetric Characterization of Oxide Thin Film

Figure 4A shows the CVs recorded in 0.1 M NaOH solution between −0.6 V and −0.1 V at a variable scan rate (0.05 Vs$^{-1}$ to 0.25 Vs$^{-1}$) with the prepared ITO-IrOx electrode substrate. The appearance of two symmetric peaks during forward and reverse scans indicate the reversible behaviour of IrOx film immobilized on the ITO surface. From the observation of XPS analysis, these reversible peaks can be assigned to Ir(III)/Ir(IV) sites on the ITO film. Here, the peak separation of the redox peaks $\Delta E_p (= E_{pa} - E_{pc})$ and the $i_{pa}/i_{pc}$ ratio were found to be 53 mV and ~1.0, respectively, for the electron transfers confined in the electrode surface. The associated peak current vs. scan rate plot is shown in Figure 4B

(data were taken from Figure 4A). The surface concentration (Γ) of IrOx film on the ITO electrode was then determined from the slope of this plot using Equation (3).

$$i_p = \frac{n^2 F^2 A \Gamma v}{4RT}$$

(3)

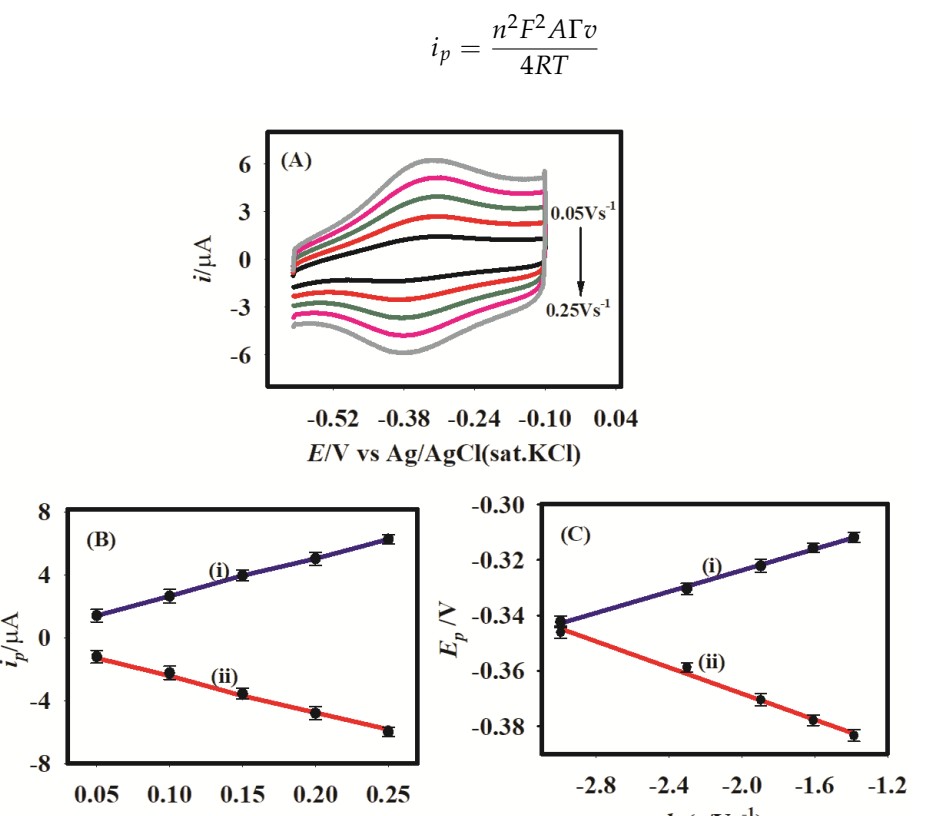

**Figure 4.** (**A**) Characterization of ITO-IrOx electrode by recording CVs in 10 mL 0.1 M NaOH at variable scan rate. (**B**) the plot of peak current ($i_p$) vs. scan rate ($v$), (**C**) plot of peak potential ($E_p$) vs. ln $v$. Where (i) and (ii) represent anodic and cathodic progress respectively.

Here, $n$ ($n$ = 1 here) is the number of electrons, $R$ = gas constant (8.314 J K$^{-1}$ mol$^{-1}$), $T$ = temperature (293 K), $A$ = electrode area (0.5 cm$^2$), $F$ = Faraday constant (96,485 C mol$^{-1}$). The active Ir site concentration (Γ) = $4.98 \times 10^{-8}$ mol cm$^{-2}$.

Conversely, Figure 4C shows plots of peak potential ($E_p$) vs. scan rate ($v$). The anodic ($E_{pa}$) and cathodic ($E_{pc}$) peak potential changed linearly as a function of ln $v$. With Laviron's Equation (4), the slope of $E_p$ vs. $ln\ v$ plot can be used to determine the transfer coefficient (α) value.

$$E_p = E° + \frac{RT}{\alpha n F} - \frac{RT}{\alpha n F} ln v$$

(4)

Here α, $T$, $n$, $F$, and $R$ indicate the cathodic electron transfer coefficient, temperature (293 K), number of electrons, Faraday constant (96,485 C mol$^{-1}$), and gas constant (8.314 J K$^{-1}$ mol$^{-1}$), respectively. From the slope of the $E_{pc}$ vs. $ln\ v$ plot, $RT/\alpha n F$ was evaluated to be 0.0236. Therefore, α was calculated to be 1.00. A very high α value indicates a reversible process as well as fast electron transfer [42]. The above observations suggest that IrOx particles were well immobilized on the ITO surface providing features of thin film electrochemistry. In a previous study, it was reported that IrO$_2$ particles immobilized on the ITO surface are capable of generating oxygen via water oxidation reactions [43]. In the present circumstance, approaches were taken to check the possibility of sensing DO using ITO-IrOx via reduction reactions.

*3.3. EIS Studies*

As per the principle of EIS, the real (Z′) and imaginary parts (Z″) of the impedance, were used to represent the complex plane plot of the EIS spectrum. The ideal Randle's circuit has a series assembly of solution resistance ($R_s$) with a parallel combination of double-layer capacitance ($C_{dl}$) and charge transfer resistance ($R_{ct}$) among other components. Before recommending modified electrode as an efficient catalytic material pertaining to ORR, it was essential to draw how the electrochemical properties of modified electrodes differ with and without the presence of DO.

Figure 5 displays the Nyquist plots of the ITO and ITO-IrOx electrode with and without the presence of oxygen at −0.85 V in 0.1 M NaOH solution. The −Z″ vs. Z′ relationship suggests that in the absence of DO, both ITO and ITO-IrOx electrodes were highly polarizable. However, in the presence of DO, the resistivity of the electrodes decreased, and least resistivity was observed for the ITO-IrOx electrode. This difference indicates that a ITO-IrOx electrode is more capable of reducing oxygen than an ITO electrode alone.

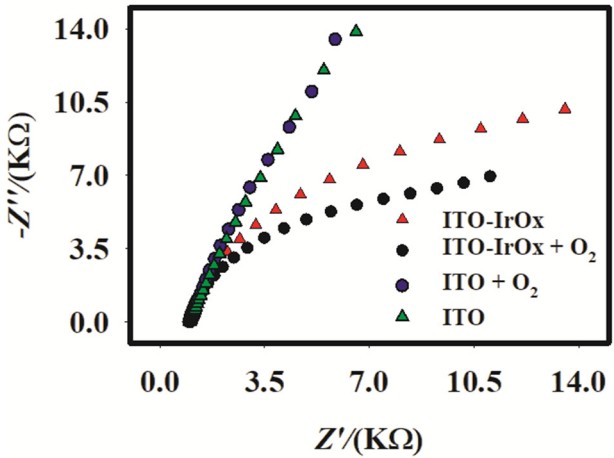

**Figure 5.** EIS spectra of ITO and ITO-IrOx with $O_2$ saturated and $N_2$ saturated in 0.1 M NaOH were recorded at −0.85 V.

*3.4. ORR Studies*

3.4.1. Catalysis

Cyclic voltammograms were recorded with ITO and ITO-IrOx substrate in 0.1 M NaOH solution with air-saturated $O_2$ at 0.2 Vs$^{-1}$ scan rate as shown in Figure 6A. An ITO does not show any well-defined peak pertaining to oxygen reduction reaction. However, the IrOx immobilized ITO electrode exhibited exclusive currents towards the reduction of dissolved oxygen. The oxygen reduction reaction involves either a two-electron transfer process generating $H_2O_2$ as an end product or a four-electron transfer process generating hydroxide ions as the final product. In this study, the evolution of $H_2O_2$ was checked by introducing a Pt-Pd catalyst to see the potential decomposition of evolved $H_2O_2$ after 2 h long bulk electrolysis of 100 mL air saturated 0.1 M solution following the procedure mentioned in the literature [44]. However, no existence of $H_2O_2$ was noticed, implying that the ITO-IrO$_x$ electrode reduced DO via a four-electron transfer process. The appearance of a well-defined cathodic peak at ca. −1.10 V, establishes the efficiency of the ITO-IrO$_x$ electrode concerning the electrocatalytic reduction of oxygen molecules dissolved in water. The effect of scan rates (recorded between 0.025 Vs$^{-1}$ and 0.5 Vs$^{-1}$) for electrochemical reduction of DO is demonstrated in Figure 6B. The gradual increase in reduction current was observed in all cases with the increase in scan rate. Figure 6C exhibits a linear fit of the peak current against the scan rate. This observation indicates a diffusion-controlled reaction ORR process that took place at the ITO-IrO$_x$ surface.

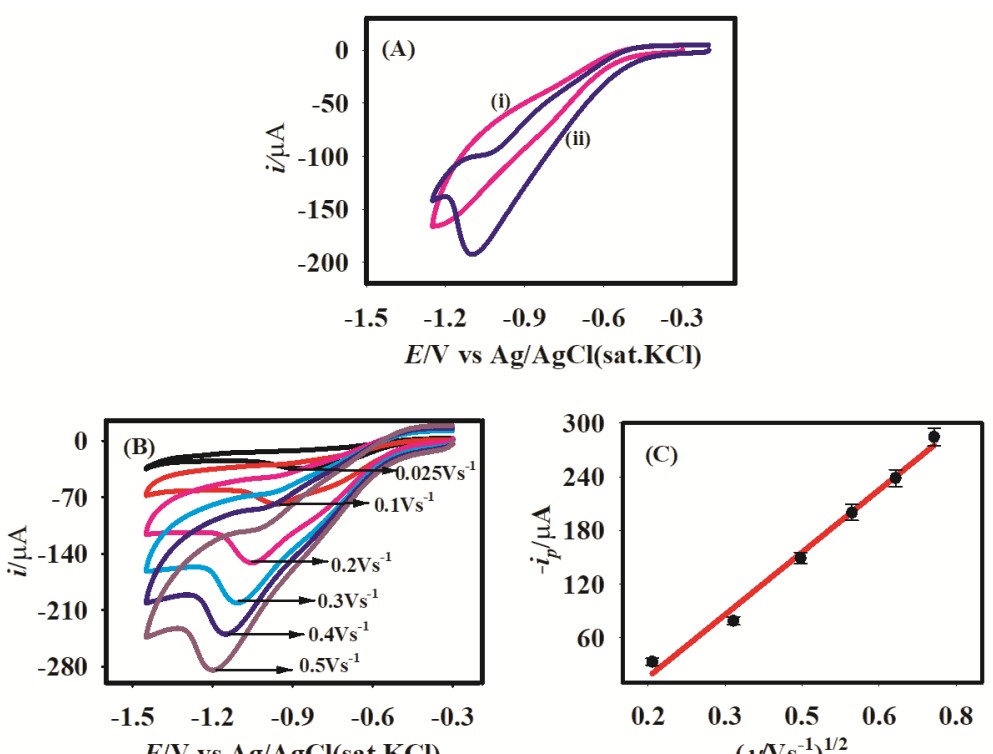

**Figure 6.** (**A**) Comparative CVs of oxygen reduction at (i) ITO and (ii) IrO$_x$-ITO electrode. (**B**) Cyclic voltammograms recorded on ITO-IrOx electrode in air-saturated 0.1 M NaOH at variable scan rates from 0.025 Vs$^{-1}$ to 0.5 Vs$^{-1}$, (**C**) $i_p$ vs. $v^{1/2}$ plot.

### 3.4.2. Sensing, Stability and Interference of Some Common Ion Study

Figure 7A illustrates the linear sweep voltammograms (LSVs) of 0.1M NaOH solution recorded for DO concentration between 0.4 ppm and 11.7 ppm using the fabricated ITO-IrO$_x$ electrode. To develop an efficient sensor for dissolved oxygen, we recorded the LSVs of DO containing water using ITO-IrOx for calculating the limit of detection (LOD) as shown in Figure 7B. It can be seen that the peak current increased linearly with the increase in O$_2$ concentrations (0 to 12 ppm). The limit of detection (LOD) of O$_2$ reduction was found as 0.49 ± 0.02 ppm by using equation 5 from triplicate experiments

$$\text{LOD} = \frac{3 \times \text{SDB}}{\text{SC}} \tag{5}$$

where, SDB is the standard deviation of the blank solution and SC is the slope of the calibration curve.

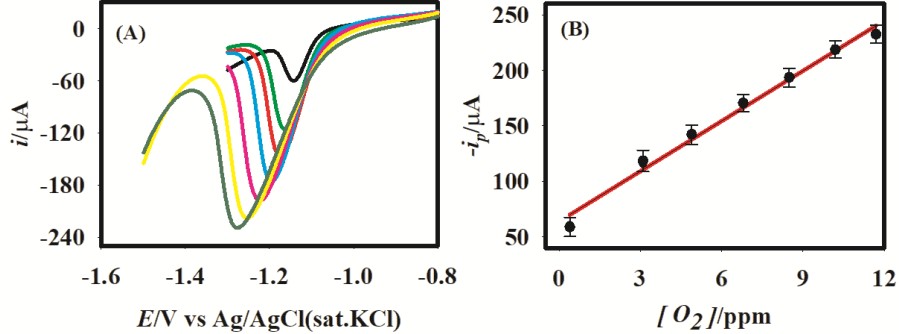

**Figure 7.** (**A**) Linear sweep voltammograms (LSV) of O$_2$ reduction on ITO-IrOx in 0.1 M NaOH solution, (**B**) a plot of peak current ($i_p$) vs. concentration of O$_2$.

The stability and interference of several ions, usually present in surface water, were checked by batch injection analysis. In a batch injection analysis process, an amperometric curve is recorded at a fixed potential. During this process, an electrolysis zone is quickly established on the electrode after the injection of a specific concentration of analyte, and if the sensor reacts with the analyte, a transient signal is generated. In the present case, a typical amperogram (*i-t* curve) was recorded at −1.1 V where different analytes (such as oxygen, nitrate ion, sulphate ion, chloride ion, carbonate ion, and methanol) were injected at the ITO-IrOx electrode surface periodically for around 90 sec (Figure 8). Here, short spikes specify the analyte other than oxygen and long spikes specify the analyte for oxygen. From Figure 8, it can be assumed that the electrode is highly stable and interference-free from several common ions. Thus, in terms of sensitivity, reproducibility, interference and stability, this sensor exhibited excellent behaviour for DO sensing.

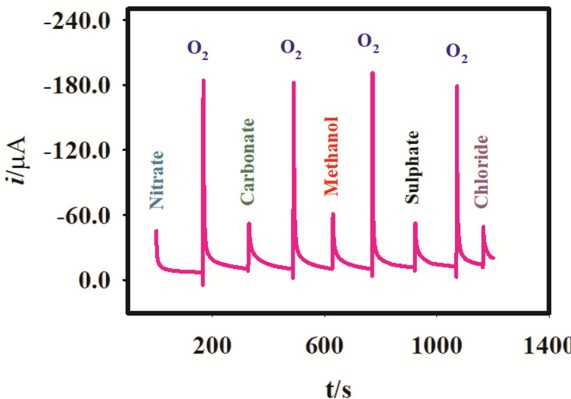

**Figure 8.** The amperometric i-t curve at potential −0.85 V of ITO-IrOx electrode −1.1 V, for checking electrode stability for $O_2$ reduction and testing different ion interference using equal concentrations as $O_2$ concentration (nitrate, carbonate, methanol, sulphate and chloride).

Finally, real sample analysis from a different source of water was performed using an amperometric method. The data are reported in Table 1.

**Table 1.** Quantification of DO in real samples with ITO-IrOx electrode.

| Sample | DO (ppm) [1] | RSD [2] (%) |
|---|---|---|
| Pond water | 6.5 | 3.2 |
| Tap water | 7.3 | 2.1 |
| Drain water | 0.9 | 10 |
| Canal water | 6.2 | 3.6 |
| Paddy land water | 5.3 | 1.4 |

[1] Mean of the three repeated determinations (S/N = 3). [2] Relative standard deviation (RSD) value indicates precision among three repeated determinations.

The results obtained for real sample analysis are consistent for DO determined with a conventional DO meter (Milwaukee 600, Rumania). Thus, the fabricated ITO-IrOx electrode could be recommended for DO measurements.

## 4. Conclusions

The present study showed an easy formation of Iridium oxide film on ITO. The band gap energy study showed that the ITO-IrOx does not support photocatalytic activity. Iridium oxide film exhibits a couple of peaks of Ir(III) and Ir(IV), which catalyse the ORR. Utilizing the modified electrode, a systematic approach has been taken to examine the sensing, interference, and kinetics of ORR. Excellent ORR activity with high stability and selectivity from interference ions (nitrate, methanol, chloride, etc.) was demonstrated at ITO-IrOx. Thus, ITO-IrOx could be employed for the dissolved oxygen sensor in the near future.

**Author Contributions:** M.S.: Writing original draft, Investigation, Formal analysis, M.M.H.: Writing-original draft, review & editing, Data curation, Surface characterizations, T.A.O.: Formal analysis, Investigation, M.A.H.: Methodology, Visualization, Writing-review & editing. All authors have read and agreed to the published version of the manuscript.

**Funding:** The authors indebtedly acknowledge financial support from the Ministry of Education, Bangladesh (PS 20201512) and Shahjalal University of Science and Technology (Grant No. PS/2021/2/02 and PS/2022/1/01) along with partial support from the Ministry of Science and Technology (Project ID: SRG-223537).

**Institutional Review Board Statement:** Not applicable.

**Informed Consent Statement:** Not applicable.

**Data Availability Statement:** Not applicable.

**Conflicts of Interest:** The authors declare no conflict of interest.

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
