# Peer review of "Electrocatalytic Reduction of O2 by ITO-IrOx: Implication for Dissolved Oxygen Sensor in the Alkaline Medium"

_2673-3293, doi:10.3390/electrochem4020012_

Round 1
Reviewer 1 Report
This work is focused on the realization of an electrochemical sensor based on the modification of an ITO electrode through electrodeposition of IrOx nanoparticles for the evaluation of the water purity. Specifically, the device works quantifying the dissolved oxygen in natural waters under alkaline conditions, by exploiting the oxygen reduction.
The work is very interesting, the electrode modification is fast end easy to make, providing for an effective improvement of the oxygen reduction promotion in comparison with the bare ITO system, as highlighted in Fig. 5A. The sensitivity, linear range, LOD of the sensor are adequate to the final application and the interferences analysis proved the high selectivity of the system. In addition, the development of the device is well characterized, therefore I really appreciated the work and I think it’s suitable for the publication in Electrochem Journal.
I have just a little suggestion:
In figure 6B, it would be appropriate to report the error bars for each value.

Author Response
Thank you for your time and suggestions. The authors updated the manuscript according to your suggestions and comments. Now, we hope that all issues are solved, and the referees would be kind to recommend the manuscript for publication in this reputed journal. Thank you again for your kind consideration.

Reviewer 2 Report
The research presented here is quite interesting. The manuscript is well presented, but the English need improvement. Especially the introduction is quite clanky and the english does not flow nicely. I recommend to edit the language in all manuscript as it need improving. Please explain the following statement: "People are constantly contaminating water with numerous types of point and non-point sources. " In row 61 and 62 change f In2O3 with the correct font with the number subscript. In section 2.2 the author need to explain in a better way what they meant by "Ethanol was added until the solution had no colour effect in addition to ethanol". This is not clear. The description of the sensor in section 2.3 will benefit form a draw or a photo of the system used. No need of this statement "Notably in this presentation, we marked the iridium oxide as IrOx as most of the literature uses the same terminology." until it is clearly stater before that IrOx mean iridium oxide. Figure 1 A and 1 B the SEM scale is not clear; so I am not sure if the images are on the same scale or not; please change them. On table one where the real samples were analysed, three samples per type of water is a really low number to stated that the sensor is working, really the bare minimum. I would recommend to repeat the test and add more samples to support your statement.
Author Response

(The authors gave the same response as above.)

Reviewer 3 Report
The title and abstract are appropriate for the content of the text. In general, the manuscript is well constructed, the experiments were well conducted, and the analysis was well performed. There are some aspects that need to be addressed before publication, therefore I recommend major revision of the following points:
1. XPS peak at 63, what does it signify?
2. Durability test of the ITO and ITO-IrOx composite electrodes in O2-saturated 1 M KOH for long period.
3. LSV plots before and after chronoamperometry studies with ITO-IrOx.
4. Comparison of the XPS peaks of Ir 2p and O 3d (inset) before and after the ORR chronoamperometry study.
5. Comparison of Nyquist plots of the ITO and ITO-IrOx composite electrodes should be plotted.
6. Comparison of the methanol tolerance for ITO-IrOx and ITO electrodes should performed by adding methanol during the chronoamperometry study.
7. LSV plots should be plotted for (b) ITO and (c) ITO-IrOx electrodes in O2-saturated 1 M KOH and 1 M KOH + 3 M methanol.
8. Koutecky–Levich (K–L) curves should be plotted at different potentials
9. Diffusion current density is very low. Please explain?
10. Calculate the no. of electron involved in this reaction from the LSV and % of H2O2 yield.
Author Response

(The authors gave the same response as above.)

Reviewer 4 Report
- The Authors should refer and comment on the paper:
Masayuki Yagi, Emi Tomita, Sayaka Sakita, Takayuki Kuwabara, and Keiji Nagai; Self-Assembly of Active IrO2 Colloid Catalyst on an ITO Electrode for Efficient Electrochemical Water Oxidation; The Journal of Physical Chemistry B; 109(46):21489-91 (2005); DOI:10.1021/jp0550208
- Authors should justify the use of 0.1 M NaOH solution in the measurements - how the measurements can be transferred and applied e.g. in surface water investigations?
- Figure 4. - In EIS interpretation the Z' and -Z" should be in the same scale (equal scale on both axis)
Author Response

(The authors gave the same response as above.)

Round 2
Reviewer 2 Report
The new version of the manuscript has improved compared to the previous one and now can be published. I recommend to check the english for minor mistakes and increase the font for the scale in figure 2 as it still unreadable.
Author Response
Thank you For your valuable comments and suggestions to improve the manuscript. The authors updated the manuscript according to your suggestions. Please check the response sheet for more information.

Reviewer 3 Report
Include more references in introduction (like Catal. Sci. Technol., 2022,12, 4727-4739, J. Mater. Chem. A, 2021, 9, 7150-7161 and https://doi.org/10.1021/acsami.3c00488 as proof that transition metal based electrocatalysts are efficient in other processes like CO2 utilization and Alcohol electrochemical oxidation reaction due to their excellent catalytic properties.
Author Response
Thank you for your valuable comments and suggestions to improve the manuscript. The authors updated the manuscript according to your suggestions. Please check the response sheet for more information.
